# Knowledge of COVID-19 Infection Guidelines among the Dental Health Care Professionals of Jazan Region, Saudi Arabia

**DOI:** 10.3390/ijerph19042034

**Published:** 2022-02-11

**Authors:** Syed Nahid Basheer, Thilla Sekar Vinothkumar, Nassreen Hassan Mohammad Albar, Mohmed Isaqali Karobari, Apathsakayan Renugalakshmi, Ahmed Bokhari, Syed Wali Peeran, Syed Ali Peeran, Loai Mohammed Alhadri, Santosh Kumar Tadakamadla

**Affiliations:** 1Department of Restorative Dental Sciences, College of Dentistry, Jazan University, Jazan 45142, Saudi Arabia; vsekar@jazanu.edu.sa (T.S.V.); nalbar@jazanu.edu.sa (N.H.M.A.); 2Conservative Dentistry Unit, School of Dental Sciences, Universiti Sains Malaysia, Health Campus, Kubang Kerian, Kota Bharu 16150, Malaysia; 3Center for Transdisciplinary Research (CFTR), Saveetha Dental College & Hospitals, Saveetha Institute of Medical and Technical Sciences University, Chennai 600077, India; 4Department of Restorative Dentistry & Endodontics, Faculty of Dentistry, University of Puthisastra, Phnom Penh 12211, Cambodia; 5Department of Preventive Dental Sciences, College of Dentistry, Jazan University, Jazan 45142, Saudi Arabia; rsakayan@jazanu.edu.sa (A.R.); abokhari@jazanu.edu.sa (A.B.); 6Department of Periodontics, Armed Forces Hospital, Jazan 82722, Saudi Arabia; doctorsyedwali@yahoo.in (S.W.P.); alipeeran@gmail.com (S.A.P.); 7Interns Affairs Unit, College of Dentistry, Jazan University, Jazan 45142, Saudi Arabia; 201310090@stu.jazanu.edu.sa; 8School of Medicine and Dentistry, Menzies Health Institute Queensland, Griffith University, Southport, QLD 4222, Australia; santoshkumar.tadakamadla@griffithuni.edu.au

**Keywords:** attitude, COVID-19, guidelines, dentistry, operative, infection control, Jazan

## Abstract

Background: This study aimed to assess the knowledge about guidelines related to COVID-19 infection control procedures among dental health care professionals (DHCPs) in the Jazan region. Methods: A cross-sectional study involving DHCPs (dental students, interns, and dentists) of the Jazan region between January and March 2021. A questionnaire with 35 items was developed and circulated online among the DHCPs. The dimensionality of the questionnaire was assessed using exploratory factor analysis (EFA). The level of awareness (LOA) was compared across the genders, level of professional experience, and exposure to guidelines. Participants were considered to have high LOA when they responded to 26 or more items correctly. Results: A total of 363 DHCPs participated in the survey. The questionnaire was found to be valid and reliable. EFA revealed a distinct three-factor structure. Moreover, 61.2% of the respondents had high LOA related to COVID-19 infection prevention. Among those who had high LOA, dentists (65.5%) were relatively more than the students (62.5%) and interns (46.2%). Among the six guideline statements related explicitly to operative dentistry, more than 50% of the respondents were aware of 3 guideline statements, while less than 50% of the respondents were aware of the remaining statements. Conclusions: Most DHCP had a high LOA for general COVID-19 infection prevention and control guidelines. Dentists, males, and those who read the guidelines had higher LOA than their counterparts.

## 1. Introduction

Coronavirus disease 2019 (COVID-19) is caused by the severe acute respiratory syndrome coronavirus 2 (SARS-CoV-2) virus, that spreads from one infected person to another. The virus can spread through the mouth, nose, or eyes in the form of droplets, aerosols, and also sometimes through contaminated surfaces [1,2]. The WHO announced COVID-19 disease as a pandemic on 11th March 2020. According to the World Health Organization [3], as of 1 October 2021, there have been 233,503,524 confirmed cases of COVID-19 globally, including 4,777,503 deaths. As of 2 October 2021, a total of 6,187,643,539 vaccine doses against COVID-19 have been administered [4]. In Saudi Arabia, the first case was reported on 2 March 2020, followed by the lockdown. Subsequently, the dental clinic reopening guidelines were released in June 2020 and updated from time to time [5].

The spread of infection in a dental office occurs either directly through droplets/aerosols or indirectly by contact with mucous membranes, saliva, respiratory fluids, and contaminated surfaces. Most dental procedures generate aerosols, especially in operative dentistry [6]. Aerosol-generating procedures (AGP) have a high potential to transmit the COVID-19 disease [7,8]. Therefore, dental health care providers (DHCP) are at high risk of exposure to SARS-CoV-2, thereby rendering them vulnerable to infection [9,10]. Although the knowledge of DHCP on COVID-19 was acceptable in the previous studies [11,12], it is imperative to ensure that all DHCP have adequate knowledge of the guidelines and their updates to protect patients and the dental team from cross-infection.

Several guidelines for COVID-19 prevention have been published by the Saudi Ministry of Health (MOH), WHO, American Dental Association, Centres for Disease Control and Prevention, and National Health Services to be adopted during the COVID-19 pandemic, which includes maintaining physical distancing, use of well-fitted masks, maintaining adequate ventilation, avoiding crowded indoor spaces, practising good hand hygiene, keeping the environment clean, covering coughs and sneezes with a bent elbow, and getting vaccinated [2,4,5,13,14]. Moreover, various schemes are recommended to reduce the spread of the virus while performing aerosol-generating procedures in a dental setting such as the nonuse of 3 in 1 syringes, the use of high-volume suction, practising four-handed dentistry, and adopting noninvasive procedures such as atraumatic restorative technique [2,5,13,15,16,17].

It is crucial for all the stakeholders to be aware of the latest guidelines and strictly implement them to mitigate the COVID-19 transmission within the dental care settings [2,4,5,13,14]. As far as the DHCP is concerned, awareness about the general guidelines and operative dentistry needs to be assessed to ensure effective implementation. Operative dentistry is vital for general dental practice because restoring the carious tooth is considered the most common treatment [18]. Awareness about all the necessary protocols, from preparing dental clinics before patient arrival until the patient leaves the dental clinic, must be thoroughly surveyed.

Recently, the DHCP belonging to university dental clinics representing four different regions (Riyadh, Jeddah, Asir and Jazan) of Saudi Arabia have been surveyed for their knowledge on COVID-19. However, the response rate was low, limiting the generalisability of their findings. The authors highlighted the positive impact of timely disseminating national guidelines to all the DHCP on their knowledge [12]. However, the awareness of DHCP about the guideline’s statements related to infection prevention and control of COVID-19 in general and during operative procedures, in particular, has not been evaluated.

Therefore, the purpose of this study was to assess the knowledge about guidelines and procedural considerations in operative dentistry while providing dental care during the COVID-19 pandemic among the DHCPs in the Jazan region of Saudi Arabia. An additional objective of the study was to explore the influence of gender, professional experience, and exposure to guidelines on the level of knowledge among DHCPs. We hypothesised that the level of knowledge among DHCPs would vary across the genders, level of professional experience and exposure to the guidelines.

## 2. Materials and Methods

### 2.1. Procedure

A cross-sectional study was conducted 10 months after the onset of the COVID-19 pandemic between January–March 2021 involving DHCP working in the Jazan region. The study was conducted according to the guidelines of the Declaration of Helsinki and approved by the Institutional Review Board of Jazan University (Ref No: CODJU-2028I). A validated questionnaire was circulated online to the participants through emails and various social media platforms.

Information about the study was widely circulated to all dental students, interns and dentists working at Jazan University through emails. An attempt was made to reach out to dentists practising throughout the Jazan region by advertising the study information on social media platforms like Facebook pages, blogs, online forums, and WhatsApp groups on dentistry. The advertising information included the link to the online survey. Those willing to participate provided consent before proceeding to complete the survey. DHCP practising in the Jazan region was only eligible to participate. DHCP who are retired, currently not practising and outside the Jazan region were ineligible to participate.

StatCalc component of the Epi info statistical program was used for sample size calculation [19]. The pilot study’s findings indicated that 50% of the subjects had high LOA. With an expected frequency of 50%, an acceptable margin of error of 5%, a confidence level of 95%, and an estimated population size of 2000, the required sample size was 322. A recent study reported 287 public sector oral health care providers in the Jazan region, but we have assumed the population size of all DHCPs in both private and public sectors to be 2000 [20].

Google Forms was used to develop the online survey, composed of three major domains: Informed consent, Demographic characteristics and COVID-19 prevention and control guidelines. The first component explained the intended purpose of the research; participants had to provide consent before answering the survey. No personal identifying information was obtained to maintain confidentiality except the email address to educate the participants with correct responses post-survey and to eliminate the data of those participants later who decided to withdraw from the study.

The face and content validity of the questionnaire was evaluated. The relevance of the questions was reviewed by two content experts from the Department of Restorative Dental Sciences who had thoroughly understood the COVID-19 prevention guidelines. After their approval, the resulting survey was pilot tested on a subset of 25 participants. Some of the items were simplified without changing the content as the participants could not comprehend them. The pilot sample constituted 25 participants (10 dental students, 6 interns and 9 dental practitioners) recruited from the Jazan University dental clinics. The pilot sample has refrained from participation in the main study. The reliability of the questionnaire on repeated administration was estimated by administering the survey to the same 25 participants after a gap of 2 weeks. After eliminating invalid responses, the internal consistency of questions was evaluated.

### 2.2. Data Collection

Demographic information (gender) and level of professional skills were recorded. Prior to questions targeting knowledge, participants’ exposure to dental guidelines and workshops on COVID-19 infection prevention and control was recorded using two items with a dichotomous response of yes/no. Knowledge was evaluated using 35 closed-ended items (Appendix A), 29 items (item 1, 3–25, 28, 32–35) were adapted from the collective clinical protocols recommended in a previous systematic review [21]. These recommendations were derived by systematically reviewing the published literature and guidelines laid down by various international healthcare institutions on general dentistry. Six questions relevant to operative dental procedures (items 2, 26, 27, 29, 30, 31) were added. These six items were developed and approved by a panel of four experts from two departments (Restorative and Preventive Dental Sciences). The questions had a five-point Likert scale ranging from ‘strongly disagree (score 1)’ to ‘strongly agree (score 5)’. Eleven questions were negatively worded (items 6, 7, 11, 18, 20, 21, 23, 27, 28, 29, 32) and were subsequently reverse scored, ‘strongly disagree (score 5)’ to ‘strongly agree (score 1)’. Overall knowledge scores were estimated by summing up the item scores, with higher scores indicating better knowledge.

Items 1–20 were related to guidelines to be followed before starting any dental procedures, while items 21–32 were related to guidelines to be followed during the dental procedures, while items 33–35 comprised guidelines to be followed after the dental procedures had been completed. It took approximately 10 min to answer the questionnaire. A participant was considered to provide a correct response when they agreed/strongly agreed to a positive statement or disagreed/strongly disagreed to a negative statement depending upon the intended meaning of the question concerning the corresponding guideline statement.

Based on the responses to the 35 knowledge items, the participant’s overall level of awareness (LOA) was assessed. The LOA was classified as high and low based on the number of correct responses; a 75th percentile was used to determine the cut-off. Participants providing 26 or more corrected responses were considered to have high LOA, while the remaining were considered to have low LOA. Awareness of each guideline statement was determined by the participants’ frequency and percentage of positive and negative responses.

### 2.3. Statistical Analysis

The collected data were analysed with the SPSS (Version 23.0; IBM, Chicago, IL, USA) software program. Descriptive statistics were conducted. Chi-square tests were performed to check the association between LOA with gender, level of professional skills, exposure to infection control guidelines and workshops. Frequencies and percentages were used to demonstrate the participants’ responses to each item.

An exploratory factor analysis (EFA) was conducted to evaluate the dimensionality of the 35-item questionnaire, principal component analysis with varimax rotation and Kaiser normalisation was used. Scree plot and Kaiser criterion (Eigenvalue > 1) were used to determine component retention. EFA was repeated by restricting the number of factors to be extracted based on the Scree plot and Kaiser criteria [22]. The Factorability of the questionnaire was determined using the Kaiser–Meyer–Olkin (KMO) test and Bartlett’s test of sphericity.

A KMO value of ≥ 0.8 with a significant Bartlett test was considered to denote an adequate sample size for factor analysis [23]. Items with factor loadings ≥ 0.30 were considered for inclusion. A cut-off of ≥0.30 is considered adequate when the sample size is over 300 [24]. Subscale scores were estimated by summing up the scores of the items in the derived factors. Internal consistency reliability of the overall questionnaire and its factors was evaluated using Cronbach’s alpha. A Cronbach’s alpha of >0.8 was considered adequate [25]. A split-half test was used to estimate reliability on repeated administration. Means and standard deviations (SD) were estimated for factor scores. Unpaired t-tests were used to compare the factor scores concerning gender and exposure of guidelines, while one way ANOVA was used for professional experience. A *p*-value of <0.05 was considered statistically significant.

## 3. Results

Overall, 61.2% of the participants had high LOA (Table 1). A greater number of dentists were found to have high LOA (65.6%) as compared to dental students (62.5%) and interns (46.2%) (*p* = 0.0001). Among 35 guideline statements, more than 50% of the participants were aware of 26 guideline statements. The majority of the respondents (54–90.9%) were unaware of nine guideline statements corresponding to items 6, 11, 21, 23, 26–29 and 32. Of these nine guideline statements, items 6 and 11 belong to the protocols to be followed before the dental procedure, and the remaining items are to be followed during the procedure. More than 50% of the participants were aware of the protocols to be followed after completing the dental procedure (Table 2).

Among the six guideline statements related to operative dentistry (item 2, 26, 28, 29, 30, 31), more than 50% of the participants were aware of items 2, 30 and 31. Furthermore, awareness of the remaining 3 guideline statements was less than 50%.

EFA demonstrated that the KMO value was 0.86, and Bartlett’s test of sphericity was significant (*p* < 0.001), indicating sampling adequacy. Three distinct factors were derived from the EFA. Factor loadings are presented in Table 3, and all the items had factor loadings of >0.3 except one item, “Any dental procedures should be delayed in patients with a history of COVID-19 for at least a month”, that had a loading of 0.28. As the item was closely related to factor 3, it was included in factor 3. The Cronbach’s value of the overall scale with 35 items was 0.81, while factors 1, 2 and 3 had a Cronbach’s alpha of 0.87, 0.85 and 0.82, respectively. Reliability on repeated administration was assessed using split-half reliability, and it was 0.91. Table 4 demonstrates that males had significantly higher scores on all the factors than females. Those who have not attended workshops had significantly higher scores for factors 2 and 3 than those who attended (*p* < 0.05).

## 4. Discussion

Documents relevant to COVID-19 infection guidelines have been released by various organisations worldwide [2,4,5,13,14]. As and when these guidance documents are periodically updated, it is the moral and ethical responsibility of the dental care health workers to be updated to prevent the spread and contain the pandemic. A survey from Turkey showed that 1.8% (*n* = 17) of the participant dentists were positively tested against COVID-19, highlighting the increased risk for dental professionals [26]. The current study was conducted to check the awareness of COVID-19 infection prevention and control guidelines recommended by various governing organisations before, during, and after performing dental procedures among dental students, interns, and dentists in the Jazan region of Saudi Arabia.

Various studies have been conducted to check the knowledge, attitudes, and practices of COVID-19 infection prevention among dental students, interns, and dentists in Saudi Arabia [12,26,27]. Stratifying the participants based on their level of professional experience was done to investigate the difference in awareness. However, we did not find any study that evaluated the awareness of a comprehensive set of COVID-19 infection prevention and control guidelines. In addition, we developed the survey adopting the questions from previous studies and evaluated the questionnaire’s validity and reliability. The questionnaire was reliable and valid, with three distinct factors demonstrating that researchers within and outside Saudi Arabia could use it.

The response rate of similar studies conducted in Saudi Arabia was 28.7% [28], 28.2% [29] and 21.7% [30]. These values are much lower than our study’s response rate of 60.5%, probably due to variation in the target population. In this study, it was found that a majority (92.29%) of the participants read the guidelines for providing dental services during the COVID-19 pandemic. There was no significant difference or association between respondents who read and did not read the guidelines of COVID-19 with LOA. This could be due to the constant update on COVID-19 infection guidelines by national and international bodies through different media.

Moreover, it was revealed in the study that only half of the participants had attended COVID-19 infection prevention workshops. However, surprisingly, fewer participants who attended the workshops had high LOA and were more aware of guidelines related to general COVID-19 cross-infection prevention and precautions in the waiting area than those who did not attend COVID-19 infection prevention workshops. This might be due to the ever-changing/updating of the guidelines as the pandemic has been evolving. DHCPs who have attended workshops might have been complacent, assuming that the workshops they have attended have provided them with comprehensive information. On the other hand, those respondents who have not participated in the workshops might constantly be making themselves aware of the evolving guidelines. It appears that the workshops conducted are not emphasising the international guidelines.

The dentists in the present study have a significantly higher level of awareness than students and interns, particularly guidelines related to preventing cross-infection in the operatory and waiting room. These findings are consistent with a study conducted among Turkish dental professionals that revealed higher knowledge about the COVID-19 aetiology, mode of transmission and the pre-procedural cautions among dental specialist respondents. These findings indicate a direct correlation between professional experience and LOA which is in total agreement with the previous studies [12,31] and disagreement with other studies reported from in Saudi Arabia [28] and Jordan [28,32]. The variation in LOA could be attributed to the difference in perception and education of the DHCP.

None of the participants in the Jazan region had poor LOA. Similar results have been reported in various studies conducted in Saudi Arabia, where the basic knowledge on COVID-19 among the DHCP was satisfactory and acceptable [12,26]. This can be attributed to the Saudi MOH’s extensive efforts to educate the DHCP about the pandemic outbreak and the associated risk of transmission. A study conducted among dental students and interns in different universities of Cairo, Egypt, revealed that they also had good knowledge and awareness about COVID-19 and the necessary precautions required to provide adequate dental treatment for the patients during the pandemic COVID-19 [33].

Among 35 guideline statements, more than 50% of participants were aware of 26 guidelines, while awareness was less than 50% for the remaining nine guideline statements. These findings are consistent with a global study that evaluated the level of knowledge and the attitude of dental practitioners related to disinfection during the COVID-19 pandemic, which indicated that the respondents did not have complete knowledge to implement disinfection guidelines specifically against COVID-19 [34]. The majority (97.8%) of participants were aware that all non-urgent conditions could be delayed by encouraging the patient to maintain proper oral hygiene. It could be attributed to the fact that the DHCP is scared and anxious in handling patients suspected of COVID-19 infection [35]. Interestingly, participants were unaware of nine guideline items (two related to the protocols to be followed before the dental procedure, and the remaining items are to be followed during the procedure). Hence, it is recommended to educate the DHCP on those guidelines before and during the dental procedure to ensure the safety and prevention of COVID-19 cross-infection.

In this study, it was alarming to know that more than 90% of the respondents were unaware of the recommendation that dental patients should be treated in rooms with negative pressure relative to the surrounding area [2,5,13,14] Similarly, 82.1% of the respondents were not aware that telephonic triage was recommended during the COVID-19 pandemic [2,5,13,14] This could be attributed to the reason that most of our survey respondents were working within university or MOH clinic sectors and not private practices. Those aspects are handled by public relations teams instead of DHCP in such facilities.

Among the items about operative dentistry, most (80%) of the respondents were unaware that during the COVID-19 pandemic, it is not recommended to use 3-in-1 syringes and ultrasonic instruments (item 29) while performing AGPs [2,5,13,15,16]. Furthermore, many (68%) respondents were unaware that four-handed dentistry (item 28) and high-volume saliva ejectors (item 26) are recommended for AGPs [2,13,16,17]. Since operative dentistry constitutes a significant part of the dental practice, the DHCP must concentrate on these guideline statements to preclude the spread of COVID-19 infection.

Based on our findings, to encourage the reading of guidelines, exams must be conducted to check the awareness of guidelines among students and interns before allowing them to work independently on patients. New evidence-based guidelines should be displayed as posters in appropriate clinical areas and posted on various social media platforms targeting the DHCPs. To increase the awareness of COVID-19 among the general masses and persuade them to follow guidelines, the government of Bhutan engaged actors, bloggers, visual artists and sports personalities [36]. Similarly, social media influencers can be hired to inform and influence the DHCPs, especially the students and interns. In addition, video resources that provide comprehensive information on the infection control procedures and sequences could be developed, similar to those developed by the Australian dental association, to demonstrate the donning and doffing sequences of the PPE [37].

The DHCPs should be reminded of the significance of doing clinical procedures in unfavourable pressure rooms, utilising large volume saliva ejectors, avoiding using 3-in-1 syringes, air-water syringes, or ultrasonic tools, and practising four-handed dentistry [2,5,13,15,16,17]. Methods such as bio-inspired systems should be adopted while performing dental procedures because these systems are showing promising results in reducing bacteremia and aerosol generation, improving immunological, microbiological, and clinical parameters [38].

This study was conducted among self-selected DHCP in the Jazan region of Saudi Arabia; therefore, the findings could not be generalised to DHCP throughout the country, and a larger sample size would yield a more precise overview of DHCP awareness. Nevertheless, the aim was to analyse the subjects in the Jazan region alone.

## 5. Conclusions

The majority of DHCP had high LOA regarding the general guidelines related to infection prevention and control of COVID-19. As far as the guidelines related to operative dentistry procedures are concerned, most DHCP was unaware that during the COVID-19 pandemic, a three-way syringe has to be avoided, four-handed dentistry should be practised, and high-volume saliva ejectors should be used during AGPs. Dentists, males, and those who read the guidelines had higher LOA than their counterparts. Based on our research findings, it is recommended to conduct lectures and seminars related to COVID-19 infection prevention control guidelines to all DHCPs in general and students and interns in particular. A task force should be organised at the institutional level to provide consolidated evidence-based guidelines and updates for all DHCPs.

## Figures and Tables

**Table 1 ijerph-19-02034-t001:** Association between levels of awareness with gender, level of professional skills, exposure of guidelines and workshops.

Variable	Level of Awareness *N* (%)		Total	Chi-Square	*p*-Value
	Low (*n* = 141)	High (*n* = 222)			
Gender					
Male	79 (36.74)	136 (63.26)	215	0.9780	0.3230
Female	62 (41.89)	86 (58.11)	148		
Level of professional skills					
Practitioners	41 (34.45)	78 (65.55)	119	33.9650	0.0001 *
Students	72 (37.50)	120 (62.50)	192		
Interns	28 (53.85)	24 (46.15)	52		
Exposure to guidelines					
Not read	15 (53.57)	13 (46.43)	28	2.8400	0.0920
Read	126 (37.61)	209 (62.39)	335		
Attendance in workshops					
Not attended	52 (29.55)	124 (70.45)	176	12.4320	0.0001 *
Attended	89 (47.59)	98 (52.41)	187		

* *p* < 0.05.

**Table 2 ijerph-19-02034-t002:** Frequency and percentage of participants’ responses to each item.

Item	Strongly Disagree *N* (%)	Disagree *N* (%)	Neutral *N* (%)	Agree *N* (%)	Strongly Agree *N* (%)
1. Patients with non-urgent conditions should be encouraged to maintain proper oral hygiene by consuming a healthy diet, avoiding hard or sticky food, and keeping good oral hygiene practices to preserve their current status.	0 (0)	0 (0)	8 (2.2)	169 (46.6)	186 (51.2)
2. Patients with reversible pulpitis and dentine hypersensitivity should be recommended analgesics if needed, avoid stimuli (cold, hot and acidic drinks or food), apply desensitising toothpaste regularly to the sensitive area with a finger, and advise the patient to call back if symptoms get worse.	6 (1.7)	31 (8.5)	61 (16.8)	155 (42.7)	110 (30.3)
3. Prevent crowding in appointment setting by booking appointments	1 (0.3)	0 (0)	23 (6.3)	120 (33.1)	219 (60.3)
4. Any dental procedures should be delayed in patients with a history of COVID-19 for at least a month	5 (1.4)	35 (9.6)	53 (14.6)	146 (40.2)	124 (34.2)
5. High-risk patients like diabetic and immunocompromised patients should be treated early in a dental office opening.	1 (0.3)	5 (1.4)	73 (20.1)	136 (37.5)	148 (40.7)
6. Telephonic triage/Tele dentistry should not be considered an alternative to in-office care.	30 (8.3)	35 (9.6)	154 (42.4)	69 (19.0)	75 (20.7)
7. Patients with fracture/loose tooth fragments or broken restorations should be referred to the designated urgent dental clinics during the COVID-19 pandemic.	11 (3.0)	28 (7.7)	93 (25.6)	137 (37.7)	94 (25.9)
8. The temperature of staff and patients should be monitored daily	2 (0.6)	2 (0.6)	16 (4.4)	70 (19.3)	273 (75.2)
9. Ask dental health care personnel to stay home if they are sick	0 (0)	2 (0.6)	17 (4.7)	91 (25.1)	253 (69.7)
10. Patients with fever should be referred to a specific medical centre treating COVID-19	8 (2.2)	7 (1.9)	31 (8.5)	128 (35.3)	189 (52.1)
11. Accompanying individuals with patients should be allowed in the clinics.	79 (21.8)	88 (24.2)	85 (23.4)	81 (22.3)	30 (8.3)
12. Hand disinfection with 60–75% alcohol should be offered upon entrance to the dental office.	2 (0.6)	1 (0.3)	33 (9.1)	149 (41.1)	178 (49.0)
13. Emergency dental care can be provided if a patient’s temperature is less than 100.4-degrees Fahrenheit and does not have symptoms consistent with COVID-19.	5 (1.4)	31 (8.5)	90 (24.8)	160 (44.1)	77 (21.2)
14. The waiting area should be large with adequate ventilation.	0 (0)	28 (7.7)	13 (3.6)	89 (24.5)	233 (64.2)
15. The 2-m separation between patients is mandatory in waiting rooms and reception areas.	0 (0)	2 (0.6)	47 (13.0)	98 (27.0)	216 (59.5)
16. Remove magazines, toys, and other objects which cannot be easily disinfected	0 (0)	29 (8.0)	44 (12.1)	84 (23.1)	206 (56.8)
17. Posters in the dental office for instructing patients on standard recommendations for respiratory hygiene/cough etiquette and social distancing should be posted in appropriate places.	0 (0)	0 (0)	29 (8.0)	160 (44.1)	174 (48.0)
18. It is not required by everyone entering the dental office to use facemasks or cloth face coverings.	133 (36.6)	81 (22.3)	41 (11.3)	77 (21.2)	31 (8.5)
19. Dental procedures require professionals to use Personal protective equipment (surgical caps, gloves, N-95 mask, FFP2 mask, goggles, gowns, and face shields).	0 (0)	6 (1.7)	39 (10.7)	112 (30.8)	206 (56.8)
20. It is not required to cover all touchable surfaces with disposable protections.	108 (29.8)	80 (22.0)	93 (25.6)	53 (14.6)	29 (8.0)
21. Patients should not be treated in rooms with negative pressure relative to the surrounding area.	12 (3.3)	21 (5.8)	119 (32.8)	160 (44.1)	51 (14.1)
22. In case hands are visibly soiled, water and soap should be used at least 20 s before using an Alcohol-based hand rub.	3 (0.8)	8 (2.2)	29 (8.0)	186 (51.2)	137 (37.7)
23. Preprocedural mouth rinse like 1.5% hydrogen peroxide or 0.2% povidone should not be used before starting any dental procedure in the patient.	40 (11.0)	54 (14.9)	120 (33.1)	79 (21.8)	70 (19.3)
24. Avoid the use of topical spray anaesthesia to prevent gag reflex	10 (2.8)	32 (8.8)	132 (36.4)	156 (43.0)	33 (9.1)
25. Use of rubber dam and N-95 masks are mandatory for aerosol-generating dental procedures	1 (0.3)	23(6.3)	47 (13.0)	124 (34.2)	168 (46.3)
26. High-volume saliva ejectors can increase aerosol or spatter while performing dental procedures	61 (16.8)	55 (15.2)	63 (17.4)	142 (39.1)	42 (11.6)
27. Panoramic radiographs or cone-beam computed tomographs should not be used for intraoral radiography	31 (8.5)	87 (24.0)	103 (28.4)	68 (18.7)	74 (20.4)
28. Four-handed dentistry should not be practised for aerosol-generating procedures.	36 (9.9)	78 (21.5)	124 (34.2)	69 (19.0)	56 (15.4)
29. Use of 3-in-1 syringes, air-water syringes, and ultrasonic instruments are allowed for all aerosol-generating dental procedures	24 (6.6)	45 (12.4)	125 (34.4)	136 (37.5)	33 (9.1)
30. Adopt the Atraumatic Restorative Technique and Chemo mechanical caries removal procedure wherever possible	0 (0)	14 (3.9)	92 (25.3)	154 (42.4)	103 (28.4)
31. To reduce the clinical time, preferences should be given to bulk-fill composite resin restorations as it permits increments up to 4 mm in thickness.	24 (6.6)	61 (16.8)	95 (26.2)	118 (32.5)	65 (18.0)
32. Treatment should be completed in multiple visits wherever possible.	60 (16.5)	68 (18.7)	83 (22.9)	119 (32.8)	33 (9.1)
33. Environmental cleaning and disinfection procedures should be followed after completion of treatment	2 (0.6)	0 (0)	45 (12.4)	124 (34.2)	192 (52.9)
34. Clean and disinfect reusable PPE	12 (3.3)	27 (7.4)	58 (16.0)	111 (30.6)	155 (42.7)
35. Manage laundry and medical waste following routine procedures	11 (3.0)	8 (2.2)	40 (11.0)	160 (44.1)	144 (39.7)

**Table 3 ijerph-19-02034-t003:** Factor loadings of the 35 items.

Item	Factor 1	Factor 2	Factor 3
	Guidelines Related to Dental Treatment Procedures	Guidelines Related to General COVID-19 Cross-Infection Control Procedures	Guidelines Related to Maintenance of Waiting Areas and Appointments/Referrals
Four-handed dentistry should not be practised for aerosol-generating procedures.	0.75		
Accompanying individuals with patients should be allowed in the clinics.	0.72		
It is not required to cover all touchable surfaces with disposable protections.	0.71		
It is not required by everyone entering the dental office to use facemasks or cloth face coverings.	0.70		
Use of 3-in-1 syringes, air-water syringes, and ultrasonic instruments are allowed for all aerosol-generating dental procedures	0.67		
Preprocedural mouth rinse like 1.5% hydrogen peroxide or 0.2% povidone should not be used before starting any dental procedure in the patient.	0.66		
Treatment should be completed in multiple visits wherever possible.	0.65		
High-volume saliva ejectors can increase aerosol or spatter while performing dental procedures.	0.65		
Patients should not be treated in rooms with negative pressure relative to the surrounding area.	0.61		
Panoramic radiographs or cone-beam computed tomographs should not be used for intraoral radiography.	0.53		
Telephonic triage/Tele dentistry should not be considered an alternate option to in-office care.	0.50		
Avoid the use of topical spray anaesthesia to prevent gag reflex	0.49		
To reduce the clinical time, preferences should be given to bulk-fill composite resin restorations as it permits increments up to 4 mm in thickness.	0.36		
Patients with reversible pulpitis and dentine hypersensitivity should be recommended analgesics if needed, avoid stimuli (cold, hot and acidic drinks or food), apply desensitising toothpaste regularly to the sensitive area with a finger, and advise the patient to call back if symptoms get worse	0.31		
Use of rubber dam and N-95 masks are mandatory for aerosol-generating dental procedures		0.72	
Environmental cleaning and disinfection procedures should be followed after completion of treatment		0.68	
The temperature of staff and patients should be monitored daily		0.66	
Adopt the Atraumatic Restorative Technique and Chemo mechanical caries removal procedure wherever possible		0.65	
Ask dental health care personnel to stay home if they are sick		0.64	
Dental procedures require professionals to use Personal protective equipment (surgical caps, gloves, N-95 mask, FFP2 mask, goggles, gowns, and face shields).		0.60	
Clean and disinfect reusable PPE		0.56	
Patients with fever should be referred to a specific medical centre treating COVID-19		0.54	
Hand disinfection with 60–75% alcohol should be offered upon entrance to the dental office.		0.50	
In case hands are visibly soiled, water and soap should be used at least 20 s before using an Alcohol-based hand rub.		0.49	
Manage laundry and medical waste following routine procedures		0.40	
The waiting area should be large with adequate ventilation			0.80
The 2-m separation between patients is mandatory in waiting rooms and reception areas.			0.72
Remove magazines, toys, and other objects which cannot be easily disinfected			0.72
High-risk patients like diabetic and immunocompromised patients should be treated in the early hours of a dental office opening.			0.65
Posters in the dental office for instructing patients on standard recommendations for respiratory hygiene/cough etiquette and social distancing should be posted in appropriate places.			0.56
Patients with non-urgent conditions should be encouraged to maintain proper oral hygiene by consuming a healthy diet, avoiding hard or sticky food, and keeping good oral hygiene practices to preserve their current status.			0.56
Patients with fracture/loose tooth fragments or broken restorations should be referred to the designated urgent dental clinics during the COVID-19 pandemic.			0.51
Emergency dental care can be provided if a patient’s temperature is less than 100.4-degrees Fahrenheit and does not have symptoms consistent with COVID-19.			0.43
Prevent crowding in appointment settings by booking appointments			0.40
Any dental procedures should be delayed in patients with a history of COVID-19 for at least a month *			0.28

* loading < 0.30.

**Table 4 ijerph-19-02034-t004:** Overall and factor scores concerning gender, level of professional skills, exposure to guidelines and workshops.

Variable		Guidelines Related to Dental Treatment Procedures	Guidelines Related to General COVID-19 Cross-Infection Control Procedures	Guidelines Related to Maintenance of Waiting Areas and Appointments/Referrals
		Mean (SD)	Mean (SD)	Mean (SD)
Gender	Males	45.64 (9.72) *	46.88 (5.95) *	41.50 (5.63) *
	Females	42.27 (9.87)	48.14 (5.02)	43.35 (4.12)
Level of professional skills	Practitioners	45.39 (10.70) †	47.47 (5.80)	41.53 (6.41) †
	Students	44.53 (9.48)	47.80 (5.02)	43.01 (3.98)
	Interns	40.73 (8.89)	45.73 (6.97)	41.15 (5.37)
Exposure to guidelines	Not read	46.64 (11.40)	44.89 (4.60) *	41.18 (4.92)
	Read	44.07 (9.76)	47.60 (5.65)	42.35 (5.16)
Attendance of workshops	Not attended	44.38 (9.70)	48.66 (4.77) *	43.60 (3.89) *
	Attended	44.16 (10.13)	46.20 (6.09)	40.99 (5.82)

* Unpaired *t*-test, *p* < 0.05; † one-way ANOVA, *p* < 0.05.

## Data Availability

The data set used in the current study will be made available at a reasonable request.

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
