# Peer review of "Knowledge of COVID-19 Infection Guidelines among the Dental Health Care Professionals of Jazan Region, Saudi Arabia"

_ijerph, 2022, doi:10.3390/ijerph19042034_

Round 1
Reviewer 1 Report
Overall, a good study. Although, assessing the awareness about the guidelines is important...I think more important would be offering some solutions based on the findings.
To conclude that they need to be reading the guidelines does not really offer anything new to the literature in my opinion. All professionals should be reading the guidelines governing their profession...if that's not being done especially in a critical circumstance such as Covid-19, how do we remedy that situation.
You could include this in the discussion with presentation of literature about how it's being done elsewhere or how awareness and reading of the guidelines might be encouraged more readily.
Author Response
Reviewer 1
We want to thank the reviewer for taking the precious time to review this manuscript and give us the comments. We would like to explicitly state that we agree with all the comments as these helped us improve the quality of our paper. We have made a conscious effort to answer all the remarks in the paper as advised by the reviewers and highlighted changes made in red for their convenience. Kindly consider these and excuse us for any lapse on our part. Please let us know if more changes need to be made to improve the paper.
Comments:
- Overall, a good study. Although, assessing the awareness about the guidelines is important...I think more important would be offering some solutions based on the findings.
Response: Thank you for your insightful suggestions and comments. We agree that solutions also need to be offered based on our observations. We included solutions in the discussion and conclusion in the revised manuscript. Line 332 to 338 and 349 to 353.
- To conclude that they need to be reading the guidelines does not really offer anything new to the literature in my opinion. All professionals should be reading the guidelines governing their profession...if that's not being done especially in a critical circumstance such as Covid-19, how do we remedy that situation.
Response: Yes, we agree that DHCPs have been advised to read the guidelines; however, since there are still some guidelines (around 9), most DHCPs are unaware of them. So, remedies in the next paragraph related to conducting workshops and developing task forces are suggested to bring about the awareness of the guidelines that have been added in the revised manuscript
3.You could include this in the discussion with presentation of literature about how it's being done elsewhere or how awareness and reading of the guidelines might be encouraged more readily.
Response: Thank you for the insightful suggestions; We added a few recommendations to increase awareness and encourage DHCPs to read and follow guidelines more readily. We have added how is the awareness increased elsewhere along with references in the discussion section of the revised manuscript. Line 320 to 331.
Reviewer 2 Report
First of all, thank you for the opportunity to review this manuscript. The aim of the following cross-sectional online survey was to assess the knowledge about guidelines related to COVID-19 infection control about operative dentistry among the dental health care professionals in the Jazan region.
The following are suggestions for the present manuscript:
TITLE:
- The title does not adequately reflect the subject of the manuscript. The authors did not examine knowledge, but attitudes.
ABSTRACT:
- The results section does not describe the most significant results. The results are better stated in the conclusion section.
- Also, the conclusions state the results, instead of synthesizing the most important result that was the goal of the research.
INTRODUCTION:
- Specify according to STROBE guidelines at the end of introduction hypotheses.
MATERIALS AND METHODS:
- Sample size calculation not done well? It was done on a pilot study, not on the total number of respondents (dental students, interns and dentists) of the Jazan region where it is conducted? How many dental students, interns and dentists are there at the Jazan region?
- The authors stated that the minimum number of respondents required should be 384. While a slightly smaller number of respondents participated in the same - 363? How do you explain that?
- Based on the pilot study, the percentage of awareness was 50% - what that mean? How were the participants of the pilot study selected? Did they later participate in the main study as well?
- As the questionnaire was designed by the author, it should have been validated first in the pilot study itself, and not later?
- On the basis of which REFERENCES the authors classified knowledge or ATTITUDES as high (26 or more correct answers; 75%), moderate (18 to 26 correct answers; 50 - 75%) and poor (less than 18 correct answers; less than 50%). Or did they decide for themselves?
- In the statistical analysis, the authors state the following: "Internal consistency reliability of the overall questionnaire and its factors was evaluated using Cronbach's alpha. A Cronbach's alpha of> 0.8 was considered adequate [20]." What is the Cronbach's alpha total questionnaire? It is not important in this paper for each individual issue, it is important in its validation, but we are only interested in the overall one.
RESULTS:
- Table 2 – unnecessary.
REFERNCES:
- Check the rules of the Journal for writing references. Abbreviated name of the Journal? Names and numbers of authors? p - pages ???
Author Response
First of all, thank you for the opportunity to review this manuscript. The aim of the following cross-sectional online survey was to assess the knowledge about guidelines related to COVID-19 infection control about operative dentistry among the dental health care professionals in the Jazan region.
Response: We want to thank the reviewer for taking the precious time to review this manuscript and give us the comments. We would like to explicitly state that we agree with all the comments as these helped us improve the quality of our paper. We have made a conscious effort to answer all the remarks in the paper as advised by the reviewers and highlighted changes made in red for their convenience. Kindly consider these and excuse us for any lapse on our part. Please let us know if more changes need to be made to improve the paper.
The following are suggestions for the present manuscript:
- Title: The title does not adequately reflect the subject of the manuscript. The authors did not examine knowledge, but attitudes.
Response: Thank you for your insightful suggestions and comments; however, we have indeed evaluated the knowledge of the DHCPs rather than the attitudes. All the items in the questionnaire evaluated the awareness of the DHCPs to each of the guidelines. Attitudes indicate what people believe is valid or essential. Therefore, we could not change the title.
- Abstract: The results section does not describe the most significant results, and the results are better stated in the conclusion section. Also, the conclusions state the results instead of synthesizing the most important result that was the goal of the research.
Response: Thank you for your insightful suggestions and comments. We apologise for the oversight; the changes have been carried out in the abstract in the revised manuscript. Line 25 to 43
- Introduction: Specify according to STROBE guidelines at the end of introduction hypotheses.
Response: Thank you for your insightful suggestions and comments. At the end of the introduction, the hypothesis has been added in the revised manuscript. Line 93 to 99
MATERIALS AND METHODS:
4.1 Sample size calculation not done well? It was done on a pilot study, not on the total number of respondents (dental students, interns and dentists) of the Jazan region where it is conducted? How many dental students, interns and dentists are there at the Jazan region?
Response: Thank you for your insightful suggestions and comments. As there were no past studies from the region, sample size calculation had to be made based on the assumptions from the pilot study. The main objective of this study was to evaluate the level of awareness among all DHCPs irrespective of their professional level. Also, the sample size of students, interns and dentists recruited was proportional to the population size of these subsets in the whole population.
We have, however, revised the sample size calculation to address your query. Line 117 to 123.
4.2 The authors stated that the minimum number of respondents required should be 384. While a slightly smaller number of respondents participated in the same - 363? How do you explain that?
Response: Thank you for your insightful suggestions and comments; we agree with your comment, and the changes have been carried out in the revised manuscript. Line 117 to 123.
4.3 Based on the pilot study, the percentage of awareness was 50% - what that mean? How were the participants of the pilot study selected? Did they later participate in the main study as well?
Response: Thank you for your insightful suggestions and comments; apologies if this was unclear. 50% of subjects from the pilot sample had a high level of awareness. The pilot sample constituted 25 participants (10 dental students, 6 interns and 9 dental practitioners) recruited from the Jazan University dental clinics. The pilot sample has refrained from participation in the main study. The corrections mentioned above have been carried out and added to the revised manuscript. Line 119 to 121 and Line 136 – 138.
4.4 As the questionnaire was designed by the author, it should have been validated first in the pilot study itself, and not later?
Response: Thank you for your insightful suggestions and comments. The questionnaire has undergone extensive psychometric evaluation; first, the face and content validity of the questionnaire was evaluated before the pilot survey. The relevance of the questions was reviewed by two content experts from the Department of Restorative Dental Sciences who had thoroughly understood the COVID-19 prevention guidelines. After their approval, the resulting survey was pilot tested on a subset of 25 participants. Some of the items were simplified without changing the content as the participants could not comprehend them. The reliability of the questionnaire on repeated administration was estimated by administering the survey to the same 25 participants after a gap of 2 weeks. After eliminating invalid responses, the internal consistency of questions was evaluated. This was followed by exploratory factor analysis, which revealed a 3-factor structure; this confirms the questionnaire validity.
The above explanation about the questionnaire validity is mentioned in the revised manuscript materials and methods section from line 131-141.
4.5 On the basis of which REFERENCES the authors classified knowledge or ATTITUDES as high (26 or more correct answers; 75%), moderate (18 to 26 correct answers; 50 - 75%) and poor (less than 18 correct answers; less than 50%). Or did they decide for themselves?
Response: Thank you for your insightful suggestions and comments. This was determined based on the 75th percentile of the correct responses; we have added clear information in the revised manuscript. Line 167 to 173.
4.6 In the statistical analysis, the authors state the following: "Internal consistency reliability of the overall questionnaire and its factors was evaluated using Cronbach's alpha. A Cronbach's alpha of> 0.8 was considered adequate [20]." What is the Cronbach's alpha total questionnaire? It is not important in this paper for each individual issue, it is important in its validation, but we are only interested in the overall one.
Response: Thank you for your insightful suggestions and comments. Corrections have been carried out, and all the reliability figures were added to the revised manuscript results. Line 222 to 225.
- Results: Table 2 – unnecessary.
Response: Thank you for your insightful suggestions and comments. Table 2 (frequency and percentage of participant's responses) is the only table where awareness /knowledge of respondents to a particular guideline/individual item is shown. So, we strongly feel that this table is of utmost necessity.
- References: Check the rules of the Journal for writing references. Abbreviated name of the Journal? Names and numbers of authors? p - pages ???
Response: Thank you for your insightful suggestions and comments. The corrections have been carried out in the revised manuscript; references have been changed to journal style.
Reviewer 3 Report
Thank you very much for allowing me to review the manuscript of considerable interest but in need of revision.Abstract lacking in materials and methods, reading the abstract does not understand the purpose of the study.
Too few and more specifics are needed to add keywords
Correct introduction, but we still need to add references to support the reduction of the spread of the virus, and what are the schemes used based on the clinical practices performed
Materials and methods, what are the primary and secondary objectives of the study? how were these participants informed? how long after the onset of the pandemic? the sample size on the basis of what were they calculated? how many other states have you considered. In addition to the declatoria of Helsinki, has the protocol been registered? Need for an Interanl review board or ethics committee document
Very confusing results, the arrangement of tables 2 and 3 are not easy to interpret, you have to reduce the text or arrange them in a different way. Re-formulate the tables to make them easier for the reader to interpret.
Discussion, I add all minimally invasive protocols to reduce the spread of the virus by aerosol, I am attaching reference: DOI 10.3390 / jcm9123914
Re-formulate the conclusion, and add teaching methods that have been implemented with new students before starting clinical practice
Author Response
Thank you very much for allowing me to review the manuscript of considerable interest but in need of revision.
Response: We want to thank the reviewer for taking the precious time to review this manuscript and give us the comments. We would like to explicitly state that we agree with all the comments as these helped us improve the quality of our paper. We have made a conscious effort to answer all the remarks in the paper as advised by the reviewers and highlighted changes made in red for their convenience. Kindly consider these and excuse us for any lapse on our part. Please let us know if more changes need to be made to improve the paper.
Comments:
- Abstract lacking in materials and methods, reading the abstract does not understand the purpose of the study
Response: Thank you for your insightful suggestions and comments. We apologise for the oversight; the changes have been carried out in the abstract in the revised manuscript. Line 25 to 43
- Too few and more specifics are needed to add keywords.
Response: Thank you for your insightful suggestions and comments; more specific keywords have been added to the revised manuscript. Line 44.
Note: dentistry, operative is a connected MeSH keyword not bound by alphabetical order. So we have used in the same order of appearance.
3.1 Correct introduction, but we still need to add references to support the reduction of the spread of the virus.
Response: Thank you for your insightful suggestions and comments; various guidelines provided by various organisations to reduce the spread of the virus has been included along with their references in the introduction in the revised manuscript. Line 66 to 72.
3.2 what are the schemes used based on the clinical practices performed
Response: Various schemes used in clinical practice to reduce the spread of the virus have been included in the introduction in the revised manuscript. Line 72 to 76.
Materials and methods
4.1 what are the primary and secondary objectives of the study?
Response: The details regarding the primary and secondary objectives and the hypotheses have been revised in the last paragraph of the introduction in the revised manuscript. Line 92 to 98.
4.2 how were these participants informed?
Response: Regarding how participants were informed regarding the study has been mentioned in the revised manuscript. Line 107 to 111.
4.3 how long after the onset of the pandemic?
Response: The study was conducted between January and March 2021, around 10 months after the onset of the pandemic. Line 101 to 102.
4.4 the sample size on the basis of what were they calculated?
Response: As there were no past studies from the region, sample size calculation had to be made based on the assumptions from the pilot study. The main objective of this study was to evaluate the level of awareness among all DHCPs irrespective of their professional level. Also, the sample size of students, interns and dentists recruited was proportional to the population size of these subsets in the whole population. We have, however, revised the sample size calculation to address your query. Line 115 to 121.
4.5 how many other states have you considered?
Response: Only the Jazan region has been considered for the current study. Line 102
4.6 In addition to the declaration of Helsinki, has the protocol been registered? Need for an Internal review board or ethics committee document
Response: Institutional review document is submitted. Line 103 to 104.
- Very confusing results, the arrangement of tables 2 and 3 are not easy to interpret, you have to reduce the text or arrange them in a different way. Re-formulate the tables to make them easier for the reader to interpret.
Response: Table 2 demonstrates the frequency and percentage of participant’s responses to each guideline/ item, and it seems self-explanatory from the legend and column headings.
Table 3 has been organised and presented according to publication standards. The table is self-explanatory, and the accompanying text interprets the results from the tables. We, however, made some minor changes to help the readers easily identify the factors/components. We are providing some articles below to demonstrate that the EFA results are presented to acceptable publication standards. For your convenience, we are only presenting COVID-19 related articles as examples below
Table 3 of https://journals.sagepub.com/doi/full/10.1177/21501327211051935
Table 2 of http://ejournal.unp.ac.id/index.php/konselor/article/view/109075
Table 2 of https://www.mdpi.com/1660-4601/18/10/5451/htm
Table 1 of https://www.frontiersin.org/articles/10.3389/fpubh.2021.787672/full
- Discussion, I add all minimally invasive protocols to reduce the spread of the virus by aerosol, I am attaching reference: DOI 10.3390 / jcm9123914
Response: Minimal invasive protocols have been included as suggestions and recommendations from the study observation, including bio-inspired systems. Reference has been added in the revised manuscript as suggested. Line 332 to 335.
- Re-formulate the conclusion, and add teaching methods that have been implemented with new students before starting clinical practice
Response: Thank you for your insightful suggestions and comments; the conclusion has been formulated, and various teaching methods have been recommended for students and interns in conclusion. Line 346 to 350.
Round 2
Reviewer 2 Report
The authors have somewhat improved the manuscript and taken into account some recommendations, but there are still shortcomings.
- The name of the used questionnaire is: "Awareness of COVID-19 infection prevention and control guidelines before, during, and after performing dental procedures among dental students, interns, and dentists in Saudi Arabia.". So please explain why the name manuscript is "Knowledge of COVID-19 Infection Guidelines among the Dental Health Care Professionals of Jazan region, Saudi Arabia ". In my opinion, this is not a knowledge questionnaire.
- Depending on the attitudes, respondents can be classified after summing up the scores obtained, into three levels of low, moderate and high attitude (Singh & Chapman, 2011 or some other authors).
Author Response
Reviewer 2:
Point 1: The name of the used questionnaire is: "Awareness of COVID-19 infection prevention and control guidelines before, during, and after performing dental procedures among dental students, interns, and dentists in Saudi Arabia.". So please explain why the name manuscript is "Knowledge of COVID-19 Infection Guidelines among the Dental Health Care Professionals of Jazan region, Saudi Arabia ". In my opinion, this is not a knowledge questionnaire.
Response: Apologies, we are having great difficulty understanding your point of view. There is a subtle difference between knowledge and awareness. By definition, Knowledge is “facts, information, and skills acquired through experience or education”. In this article, we assess the knowledge of the DHCPs that they would have gained through their experience and exposure to COVID-19 guidelines. On the other hand, awareness is a broader concept, it is “Perceiving, knowing, feeling, or being conscious of events, objects, thoughts, emotions, or sensory patterns”. We have revised the questionnaire to “knowledge…..” According to our previous response, we do not understand by any means how the items in the questionnaire could relate to assessing the attitudes. We are, however, happy to consider any logical arguments/suggestions from the reviewer, supported by evidence.
Point 2: Depending on the attitudes, respondents can be classified after summing up the scores obtained, into three levels of low, moderate and high attitude (Singh & Chapman, 2011 or some other authors).
Response: We have categorised individuals based on 75% percentile which is scientific and logical. The questionnaire we used was customised instrument and hasn’t been used by any other article. Any classification based on previous articles, for eg.,Singh and Chapman (Knowledge, attitudes and practices on disposal of sharp waste….) is not applicable to our study as we have completely different questionnaire and scoring criteria. In addition to classification based on percentile, we used advanced statistical approach to evaluate the validity of the questionnaire.
Reviewer 3 Report
The manuscript was revised following all my comments
Author Response
Comment: The manuscript was revised following all my comments
Response: We want to thank the reviewer for taking the precious time to review this manuscript and give us the comments. We would like to explicitly state that we agree with all the comments as these helped us improve the quality of our paper.